# The Impacts of a COVID-19 Related Lockdown (and Reopening Phases) on Time Use and Mobility for Activities in Austria—Results from a Multi-Wave Combined Survey

Lukas Hartwig * , Reinhard Hössinger, Yusak Octavius Susilo and Astrid Gühnemann

University of Natural Resources and Life Sciences, Vienna, Department of Landscape, Spatial and Infrastructure Sciences, Institute for Transport Studies, Peter Jordan-Straße 82, 1190 Vienna, Austria; reinhard.hoessinger@boku.ac.at (R.H.); yusak.susilo@boku.ac.at (Y.O.S.); astrid.guehnemann@boku.ac.at (A.G.)
* Correspondence: lukas.hartwig@boku.ac.at; Tel.: +43-1-47-654/856-15

**Abstract:** When activity locations were shut down in the first lockdown to prevent the spread of COVID-19 in Austria, people reduced their trips accordingly. Based on a dataset obtained through a weeklong mobility and activity survey we analyse mobility and time use changes, as well as changes in activity locations and secondary activities. Regression analysis is used to analyse differences in time use changes between socio-demographic groups. We show that trip rates and distances as well as public transport use dropped significantly during the lockdown and did not recover fully in the subsequent opening phase. Former travel time was used for additional leisure, sleep, domestic tasks, and eating in the lockdown, but only the latter two retained their increases in the opening phase. The lockdown resulted in a convergence of time use of socio-demographic groups with formerly different patterns, but the differences reappeared in the opening phase. Our findings are consistent with results from the literature but offer an integrated perspective on mobility and time use not found in either mobility- or time use-focussed studies. We conclude that there is a potential for trip reduction through a shift to virtual performance of activities, but the extent of this shift in post-pandemic times remains unclear.

**Keywords:** COVID-19; time use; mobility; virtual activities; secondary activities

## 1. Introduction

While mobility is an essential part of performing social and economic activities, the current transport system—shaped over the last decades by greatly increasing motorisation—produces severe negative consequences for the environment and society. Impacts such as air pollution, noise, greenhouse gas emissions, or inequities in access to essential services frequently exceed limits beyond environmentally and socially acceptable standards. Understanding how mobility-related activities affect travel and how and to what extent individuals are able to modify their behaviour, e.g., during disrupted periods, is crucial for creating a sustainable, inclusive, and resilient transport system in the future. Travel has long been understood as a demand predominantly derived from the demand for activities [1], resulting from accessing the locations where these activities can be performed under temporal, social, and budgetary constraints [2]. The spatial and temporal distribution of activities is therefore shaping mobility behaviour and, correspondingly, when most activities were restricted and locations were shut down during lockdowns due to the global COVID-19 pandemic, transport volumes dropped as expected. With transport volumes, greenhouse gas emissions also dropped in the affected countries, even though this effect was mostly limited to lockdown periods [3–5]. Several studies have analysed the impacts of the pandemic on traffic volumes and passenger numbers based on movement data such as traffic counts [6,7], mobile phone data [8,9], or floating car/routing data [10–12]. Other studies use surveys on reference trips [13], ask about trip frequency directly [14], or asked

for participants' changes on Likert scales [15,16] in order to analyse the decreases in trip rates, distances, and choice of transport modes. One study uses a travel activity survey, which asked participants to recall their travel before the pandemic emerged [17]. Previous studies have focussed either on changes in one specific mode like public transport [18] or cycling [6], or researched shifts between multiple modes [7,15,19,20]. Besides the forced reduction in trip rates, most of the studies also observe a modal shift away from public transport [18] towards the private car and non-motorised modes [7,10,13,15,21] which confirms the observations from traffic data.

There are some studies from time use research about the changes brought about by COVID-19 lockdowns [22,23] which observe a shift from time spent on travelling and employment towards more domestic and leisure activities. These studies have their focus on the performed activities but lack detail about mobility information such as availability of mobility tools and trip attributes. The pilot of a combined mobility and activity diary survey based on tracking and activity sensing with the use of a smartphone app looks promising [24]. Due to the nature of the pilot, the sample size was small, and thus the results from the test period which included different phases of COVID-19 restrictions in Singapore were not analysed in-depth.

Furthermore, most previous studies examine the impacts of COVID-19 based on cross-sectional observation and fail to capture the effects of weekly behaviour patterns or potential longer-term behavioural adaptations. One exception is the long-term MobisCOVID panel [25] where participants tracked their trips before as well as for up to six months after the beginning of the pandemic. Still, a research gap remains with regard to medium- to long-term behaviour change in different contexts and the interplay of activity performance and mobility. Further knowledge about these effects would provide a better understanding of the potential for long-term behaviour changes such as a reduction in travel demand towards more sustainable transport systems. It is also crucial to learn how to provide mobility services that inclusively ensure access even in crisis situations.

In this paper, we present the results from the analysis of a multi-day and multi-wave time use and mobility survey carried out before, during, and after the first COVID-19 related lockdown in Austria. We investigate how the lockdown affected the location of activities (at home/out-of-home), the time spent on them, and the engagement in secondary activities (activities performed simultaneously with the main activity through multitasking). We are especially interested in how socio-demographic factors moderated the observed effects. Finally, we discuss the implications of our findings for a transition to a more sustainable transport system.

## 2. Materials and Methods

### 2.1. Study Design

The data used for this paper originate from a conjoined Mobility-Activity-Expenditure-Diary (MAED) survey [26] conducted from September 2019 to August 2020 in Austria. This type of survey was chosen because it allows the concurrent collection of both trips and activities with their corresponding attributes such as locations, start and end times, chosen modes, activity categories, and secondary activities. All this information is reported in a diary over a continuous period of seven days, i.e., a typical work-leisure cycle. It combines the elements of well-established travel behaviour surveys/diaries [27–29] and time use diaries [30,31]. Additionally, information about available mobility tools, household characteristics, employment, income, and further socio-demographic data were collected.

In order to obtain the expenditure data in the best possible quality, we utilised a large consumer expenditure survey (CES) conducted by the national statistical office Statistics Austria from June 2019 to May 2020, and conducted a mobility-activity survey (MAS) with a subgroup of CES-participants. MAS participants received support calls and reminders to help with filling out the questionnaire. A EUR 25 voucher was provided for each participant on completion of both the survey and validation process. Trained interviewers provided support to the participants via telephone and email and checked the questionnaires for

plausibility on completion. The response rate calculated as the share of participants who completed the CES as well as our survey according to RR1 in [32] is 16.0 percent.

## 2.2. Sampling

The net sample includes 908 participants representative of the Austrian population (a detailed sample description can be found in Tables A3 and A4 in the Appendix A); it represents a continuous diary-based record of trips and activities for 168 h from each participant. The survey was conducted between autumn 2019 and summer 2020. The COVID-19 pandemic reached Austria during this period. A lockdown that was put in place from 16 March to 30 April 2020 drastically influenced trips and activity patterns of the participants, because travel continued to be allowed only for urgent private or professional reasons, as well as for necessities such as grocery shopping and use of public spaces alone or with household members [33]. Schools were closed on 16 March for senior grades and on 18 March for all other grades and remained closed until 4 May (senior year students), 18 May (lower grades), and 3 June (all other senior grades) [34].

Based on this lockdown, we divided the sample into three groups: The first group c0 ('no COVID') consists of participants from the phase that took place before any COVID-19 related restrictions existed, that is, from 18 September 2019 (first reporting day) until 15 March 2020 (last reporting day). The second group c1L ('COVID + lockdown') contains participants between 16 March to 30 April 2020 while the COVID-19 lockdown was in place. Group c1O ('COVID + opening phase') includes participants between 1 May and 14 August 2020 during the opening phase after the lockdown.

For survey participation both an online and a paper-based questionnaire were available. In c0 only 28.6 percent of participants chose online. In c1L and c1O, we promoted online participation in order to save effort for data entry of paper questionnaires, which increased the share of online participants to 68.7 percent for these phases. This change in the type of participation did not result in lower data quality or systematic exclusion of certain groups, as there is no statistically significant difference regarding age, gender, or employment between the phases c0 and c1L or c1O, respectively. The survey questions and activity categories can be found in the Appendix A in Tables A1 and A2.

## 2.3. Data Analysis Methods

In the first step, the data have been analysed descriptively, focussing on the changes in time use for various activities, the share of activities performed at home, and secondary activities. In a second step, linear regression with interaction terms was used to reveal the moderating influence of socio-demographic variables on these changes. The interaction terms were formed as a product of the respective socio-demographic variable with the dummy of the survey phase (c0, c1L, c1O). The t-value of the corresponding parameter was used to test the interaction effects for statistical significance. The results, presented further below in Tables 1 and 2, include only interaction effects with $|t| > 1.96$ corresponding to a significance level of $\alpha < 0.05$.

**Table 1.** Results of linear regression with a COVID episode dummy as the main effect and different moderator variables with significant interaction effect, comparing the situation before the lockdown c0 with the lockdown episode c1L.

| Dependent Variable (y) | $\frac{\text{Mean}(y_{c1L})}{\text{Mean}(y_{c0})}$ | Moderator Variable (z) | $\beta_z$ | $\beta_{cz}$ | t-val. ($\beta_{cz}$) |
|---|---|---|---|---|---|
| Travel [†] | 0.25 | Age | −0.19 | 0.12 | 3.59 |
| | | University educated [a] | 0.18 | −0.13 | −3.87 |
| | | Employed | 0.23 | −0.07 | −2.23 |
| | | Household (hh) urbanity [b] | 0.26 | −0.12 | −3.83 |
| | | Max workplace flexibility [c] | 0.17 | −0.10 | −2.95 |
| | | Local PT season ticket owned | 0.23 | −0.14 | −4.35 |

**Table 1.** *Cont.*

| Dependent Variable (y) | $\frac{\text{Mean}(y_{c1L})}{\text{Mean}(y_{c0})}$ | Moderator Variable (z) | $\beta_z$ | $\beta_{cz}$ | t-val. ($\beta_{cz}$) |
|---|---|---|---|---|---|
| Sleep [†] | 1.03 | Age | 0.11 | −0.10 | −2.46 |
| | | Local PT season ticket owned | 0.01 | 0.08 | 1.98 |
| | | Daily working time | −0.32 | 0.09 | 2.47 |
| Eating [†] | 1.27 | Nº of persons under 16 in hh | 0.02 | 0.17 | 4.31 |
| Employment [†] | 0.77 | Gender female [d] | −0.21 | 0.10 | 2.57 |
| | | Nº of persons under 16 in hh | 0.14 | −0.08 | −2.09 |
| | | Daily working time | 0.81 | −0.06 | −2.53 |
| Education [†] | - | - | - | - | - |
| Personal [†] | - | - | - | - | - |
| Domestic [†] | 1.10 | Gender female | 0.29 | −0.08 | −1.98 |
| Shopping [†] | 0.64 | Gender female | 0.15 | −0.10 | −2.57 |
| Leisure [†] | 1.29 | Age | 0.14 | −0.08 | −2.12 |
| Mode share PT | 0.18 | Age | −0.22 | 0.12 | 3.15 |
| | | University educated | 0.15 | −0.13 | −3.25 |
| | | Household urbanity | 0.36 | −0.19 | −5.17 |
| | | Car availability [e] | −0.38 | 0.16 | 4.54 |
| | | Local PT season ticket owned | 0.61 | −0.32 | −11.16 |
| Trip frequency | 0.36 | Age | −0.11 | 0.08 | 2.41 |
| | | University educated | 0.16 | −0.11 | −3.35 |
| | | Max workplace flexibility | 0.13 | −0.08 | −2.54 |
| | | Train discount card owned | 0.12 | −0.10 | −2.96 |
| | | Local PT season ticket owned | 0.08 | −0.10 | −2.97 |

Values in grey for $\beta_z$ are not statistically significant at $\alpha < 0.05$. [†] Time spent on the respective activity, in minutes per week. [a] Dummy: Highest completed education is university: 0 = no, 1 = yes. [b] Classification of the urbanity of the household location: 1 = rural, 2 = intermediate, 3 = urban. [c] Share of work that can be done from a self-chosen location: 1 = (almost) none, 5 = (almost) all. [d] Dummy: 0 = male, 1 = female. [e] Personal availability of a private car: 1 = never, 2 = sometimes, 3 = always.

**Table 2.** Results of linear regression with a COVID episode dummy as the main effect and different moderator variables with significant interaction effect, comparing the situation before the lockdown c0 with the opening phase c1O.

| Dependent Variable (y) | $\frac{\text{Mean}(y_{c1O})}{\text{Mean}(y_{c0})}$ | Moderator Variable (z) | $\beta_z$ | $\beta_{cz}$ | t-val. ($\beta_{cz}$) |
|---|---|---|---|---|---|
| Travel [†] | 0.76 | Age | −0.21 | 0.07 | 2.10 |
| | | University educated [a] | 0.21 | −0.08 | −2.17 |
| | | Employed | 0.22 | −0.09 | −2.43 |
| | | Household (hh) urbanity [b] | 0.30 | −0.07 | −2.06 |
| | | Max workplace flexibility [c] | 0.16 | −0.10 | −2.73 |
| | | Local PT season ticket owned | 0.27 | −0.08 | −2.44 |
| | | Daily working time | 0.17 | −0.12 | −3.42 |
| Sleep [†] | 1.01 | Age | 0.09 | −0.10 | −2.68 |
| | | Employed | −0.31 | 0.08 | 2.36 |
| | | Gender female [d] | 0.10 | −0.09 | −2.39 |
| | | Physical activity at job [e] | −0.17 | 0.08 | 2.30 |
| | | Driver's license owned | −0.17 | 0.08 | 2.28 |
| | | Car availability [f] | −0.14 | 0.09 | 2.43 |
| | | Total nº of persons in hh | −0.01 | 0.07 | 1.98 |
| | | Daily working time | −0.30 | 0.12 | 3.56 |
| Eating [†] | - | - | - | - | - |
| Employment [†] | 0.88 | Personal income | 0.41 | −0.07 | −2.14 |
| | | Max workplace flexibility | 0.54 | −0.09 | −2.90 |
| | | Daily working time | 0.77 | −0.19 | −8.44 |
| Education [†] | 0.90 | University educated | 0.05 | 0.08 | 2.05 |
| Personal [†] | 0.82 | Personal income | −0.08 | −0.10 | −2.78 |

**Table 2.** *Cont.*

| Dependent Variable (y) | $\frac{\text{Mean}(y_{c1O})}{\text{Mean}(y_{c0})}$ | Moderator Variable (z) | $\beta_z$ | $\beta_{cz}$ | t-val. ($\beta_{cz}$) |
|---|---|---|---|---|---|
| Domestic [†] | 1.09 | Personal income | −0.16 | 0.09 | 2.61 |
|  |  | Physical activity at job | −0.19 | 0.08 | 2.17 |
|  |  | Daily working time | −0.31 | 0.12 | 3.32 |
| Shopping [†] | 1.00 | Driver's license owned | 0.00 | −0.07 | −2.02 |
| Leisure [†] | 1.08 | Daily working time | −0.22 | 0.08 | 2.13 |
| Mode share PT | 0.66 | Age | −0.23 | 0.08 | 2.15 |
|  |  | University educated | 0.12 | −0.12 | −3.40 |
|  |  | Household urbanity | 0.41 | −0.08 | −2.34 |
|  |  | Max workplace flexibility | 0.03 | −0.09 | −2.50 |
|  |  | Local PT season ticket owned | 0.67 | −0.16 | −5.93 |
| Trip frequency | 0.79 | Daily working time | 0.15 | −0.08 | −2.30 |

Values in grey for $\beta_z$ are not statistically significant at $\alpha < 0.05$. [†] Time spent on the respective activity, in minutes per week. [a] Dummy: Highest completed education is university: 0 = no, 1 = yes. [b] Classification of the urbanity of the household location: 1 = rural, 2 = intermediate, 3 = urban. [c] Share of work that can be done from a self-chosen location: 1 = (almost) none, 5 = (almost) all. [d] Dummy: 0 = male, 1 = female. [e] Share of tasks with high level of physical activity at the job: 1 = (almost) none, 5 = (almost) all. [f] Personal availability of a private car: 1 = never, 2 = sometimes, 3 = always.

## 3. Results

### 3.1. Description of the Sample

Tables A3 and A4 in the Appendix A describe the sample based on socio-demographic characteristics of household and person level. By comparing the distribution of socio-demographic variables with that of the total Austrian population from the national statistics office Statistics Austria [35], the representativeness of our sample can be evaluated.

Generally, the net sample of 908 persons from 551 households can be seen as representative. Noteworthy differences exist with respect to household size. Single-person households and those with more than four persons are both underrepresented. Single-person households are often found in cities, so households from predominantly urban federal states (Vienna and Salzburg) are also slightly underrepresented, as can be seen in Figure 1e.

Participants not in employment are underrepresented (Figure 1c), and participants with high levels of education (Figure 1d) as well as female participants are overrepresented (Figure 1a). With respect to the age, both the youngest and oldest age groups are slightly underrepresented (Figure 1b).

A comparison of the different survey phases against each other shows some deviations of the lockdown phase c1L. Employed persons and school-age participants (16–19) participated more frequently, whereas urban inhabitants, self-employed persons, and those aged 50–59 participated less frequently.

### 3.2. Mobility Indicators

The mobility indicators in our sample corroborate the findings from the literature. Trip rates went down from previously 2.9 to 1.1 trips/day during the lockdown and rose to 2.3 trips/day in the opening phase. Travel distances per trip also declined drastically from an average of 13.8 to 7.8 km/trip during the lockdown, before bouncing back to 12.4 km/trip in the subsequent opening phase.

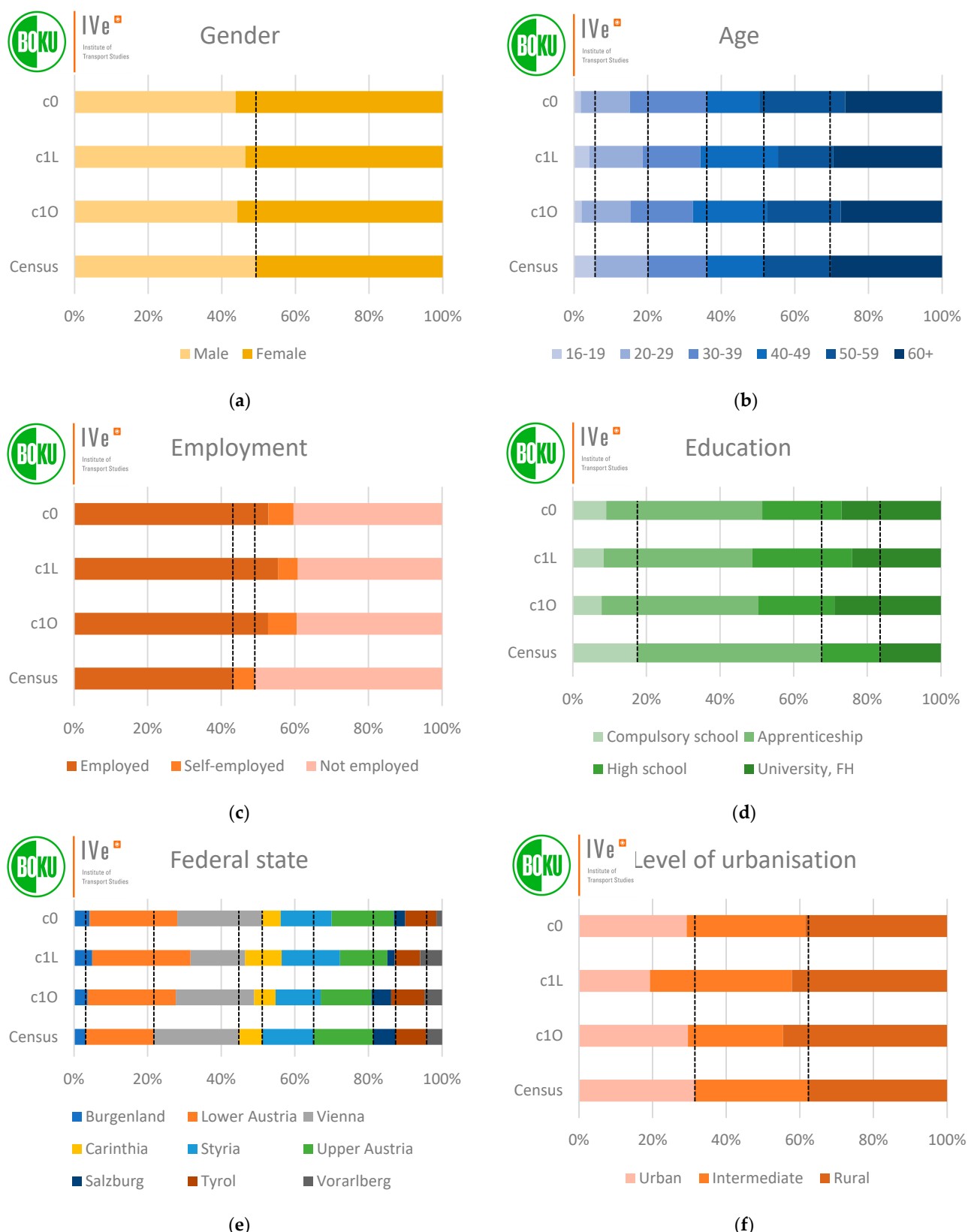

**Figure 1.** Composition of the different waves of the study sample in comparison with national census data with respect to (**a**) Gender, (**b**) Age, (**c**) Employment, (**d**) Education, (**e**) Federal state, and (**f**) Level of urbanisation.

Figure 2 shows how the average weekly number of trips per person changed for different modes. In *c1L* trips with most modes went down drastically, but car trips as a driver were reduced much less. Public transport trips were reduced substantially and continued to be so in *c1O*. This effects a modal shift away from public transport mainly to driving. The sharp drop in the number of public transport trips by 96 percent and the drop in public transport's mode share by 11 percent as well as the increase in the mode share of driving by 16 percent during the lockdown despite a reduction of car trips by 53 percent are comparable to findings in other studies [7,13,18,36,37]. This makes sense, given that the inability to socially distance in public transport constituted an increased risk for infection [38]. A modal shift towards walking and cycling that is reported in many studies, e.g., [13,15], manifests in our dataset only in the opening phase after the lockdown. This is probably caused by methodological differences: our participants were instructed to report purely recreational trips without a specific destination (walk the dog etc.) not as 'trip' but as leisure activity because the data are used for time use models, which require a strict distinction between trips to reach a destination and those that are an end in themselves.

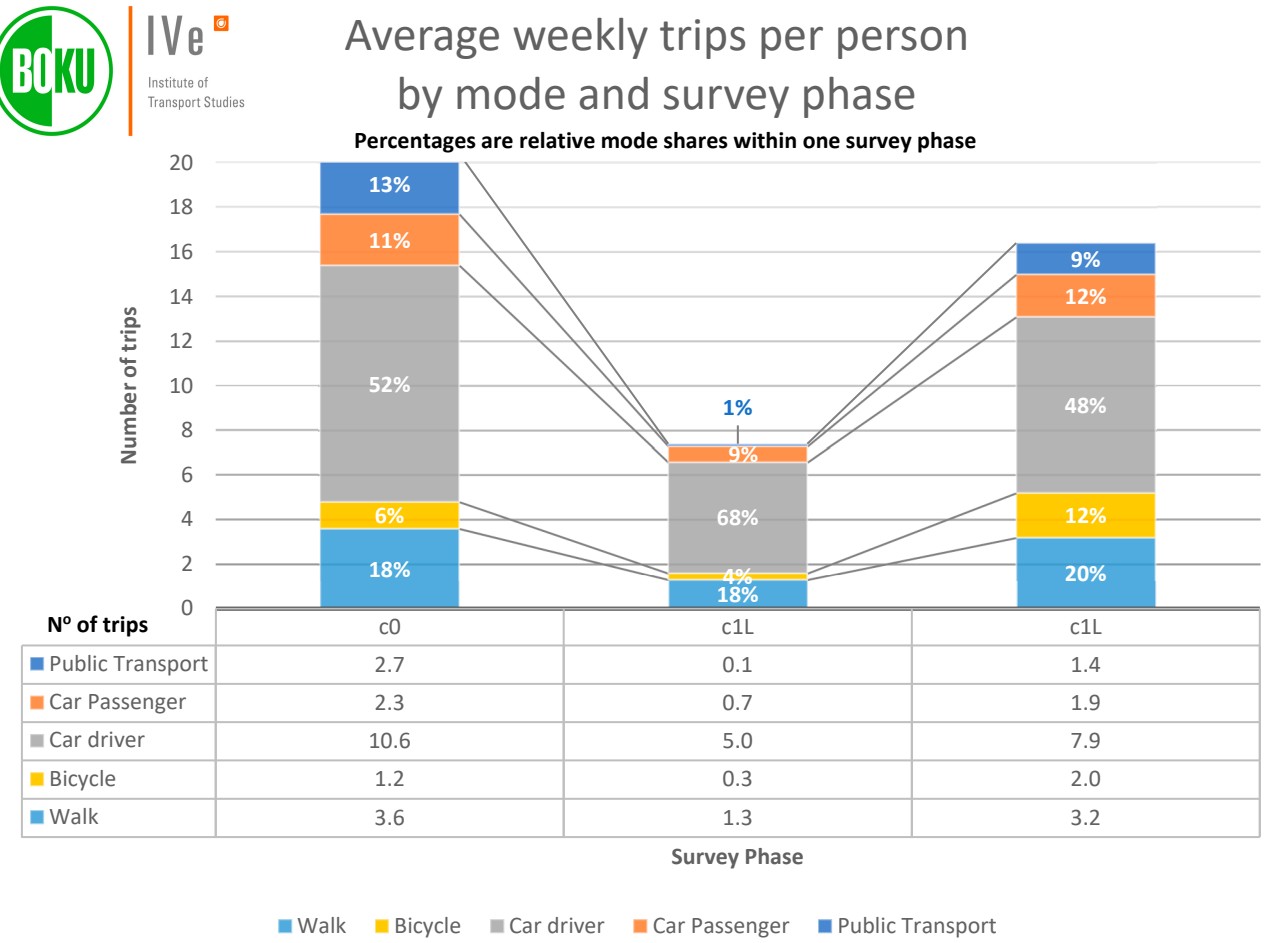

**Figure 2.** Average weekly number of trips per person by mode and survey phase.

The remaining parts of this paper are about the changes in time use, i.e., the time spent on different activities, and the location where the activities are performed. A better understanding of these changes can help to explain the observed change in mobility.

### 3.3. Share of in Activities Performed at Home

Since the location of activities determines travel behaviour, our first analysis of time use looks at the changes where activities were carried out. Based on the known home address of the respondents and the reported destination of each trip we can identify whether

an activity has been carried out at home or out of home. The share of activities carried out at home is shown in Figure 3. During the lockdown phase, the share of activities that were performed at home increased for all activities, compared to the situation before the lockdown (Δ $c1L$ in Figure 4). As can be seen in Figure 4, the highest increases were for education, employed work, leisure, and eating. Shopping increased only very little and from a low level. We do not see the estimated share of online shopping of around 13 percent in Austria [39] in our data and other studies found the COVID-19 pandemic to be a driver for online shopping [40,41]. It should, however, be noted that 'shopping' as an activity takes very little time compared to, e.g., sleeping, paid work, domestic work, and leisure. In particular, online shopping, which is done next to other digital activities, is therefore likely underreported. Hence, both values for the change in shopping location during lockdown and during the opening phase compared to c0 (Δ $c1O$ and Δ $c1O$ in Figure 4) are too small to be statistically significant and should not be interpreted.

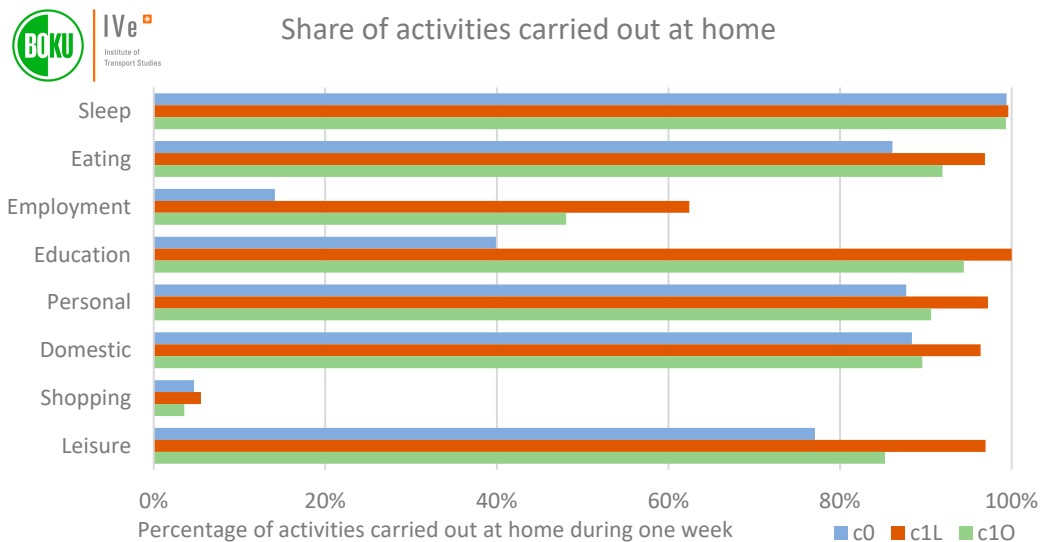

**Figure 3.** Share of activities carried out at home by activity category and survey phase.

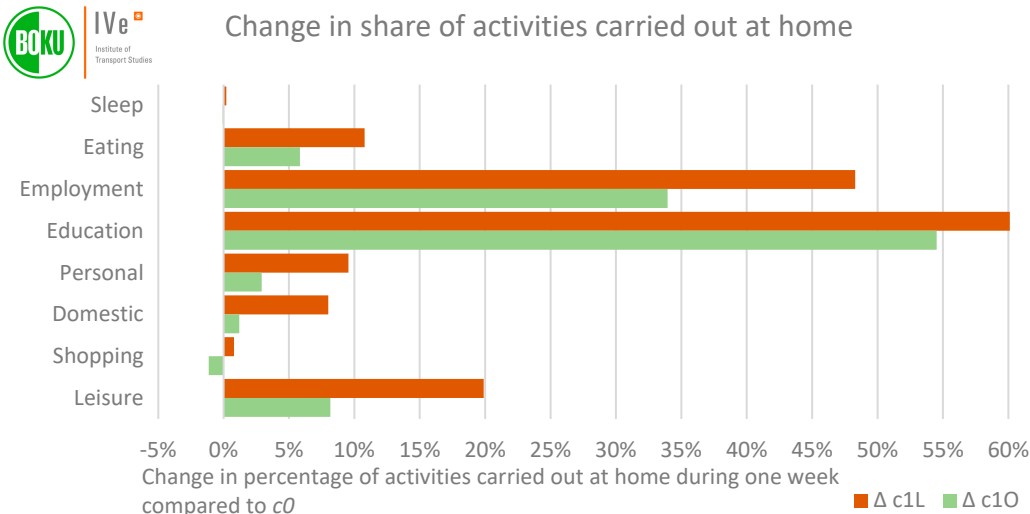

**Figure 4.** Change in share of activities carried out at home compared to situation before the lockdown by activity category and lockdown phase.

When the lockdown was over, the at-home shares for personal and leisure activities quickly dropped back close to before-lockdown levels in the opening phase c1O. The at-home share of employed work also decreased from 62.4 to 54.5 percent in c1O. Contrastingly,

the share of education activities conducted at home remained high at 94.4 percent, which is very different compared to the situation in c0 (Δ c1O in Figure 4). This is a specific situation in Austria, where schools and universities kept (or went back to) distance learning much longer than in other countries.

### 3.4. Secondary Activities

Before the lockdown (c0), 30.9 percent of all activities were performed with a secondary activity (see Figures 5 and 6). This share increased in the lockdown phase c1L to 37.2 percent and dropped back to 35.6 percent in c1O. The increase in secondary activities in c1L was biggest for leisure as well as employed work, eating, and domestic tasks. After the lockdown, secondary activities for leisure subsided, whereas those for eating, employed work, and domestic remained high or increased further.

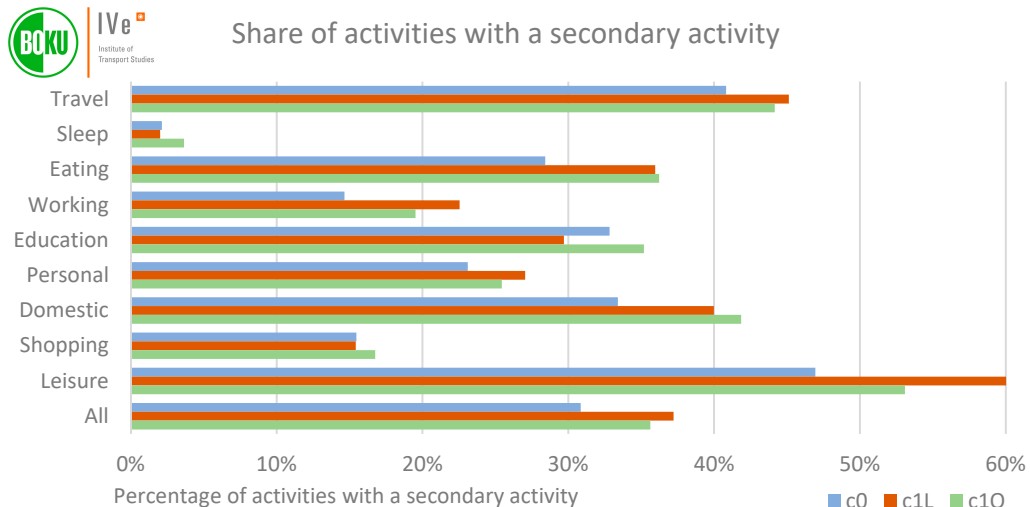

**Figure 5.** Share of activities with a secondary activity.

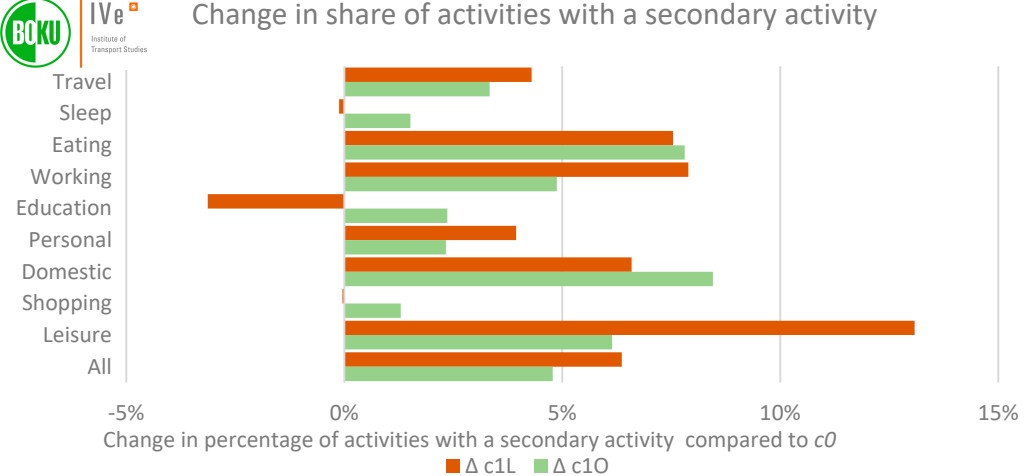

**Figure 6.** Change in share of activities with a secondary activity compared to the situation before the lockdown.

This increase is mostly due to the increase in secondary activities at home in c1L and c1O. In c0 the share of activities with a secondary activity at home was 27.3 percent and 29.5 percent for out-of-home activities. In c1L significantly more at home secondary activities were performed (37.4 percent) while the share of out-of-home secondary activities is 24.1 percent. In c1O the split is nearly even (34.0 percent at home; 32.4 percent out-of-home).

### 3.5. Shift in Time Use for Main Activities

The change in time use due to the lockdown for various main activities is presented in Figures 7 and 8. It can be seen that the time spent for travelling dropped dramatically during the lockdown and was still down nearly 25 percent in the opening phase compared to the situation before.

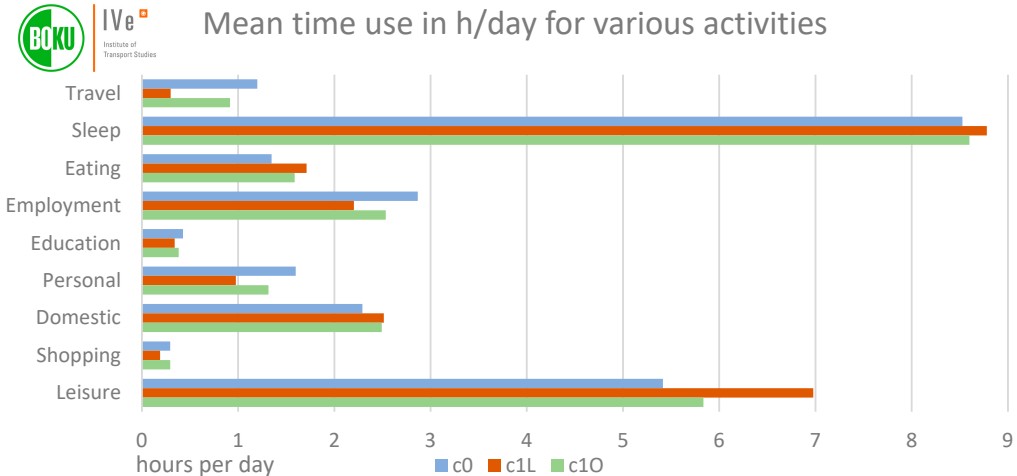

**Figure 7.** Mean time use in hours per day for various activities.

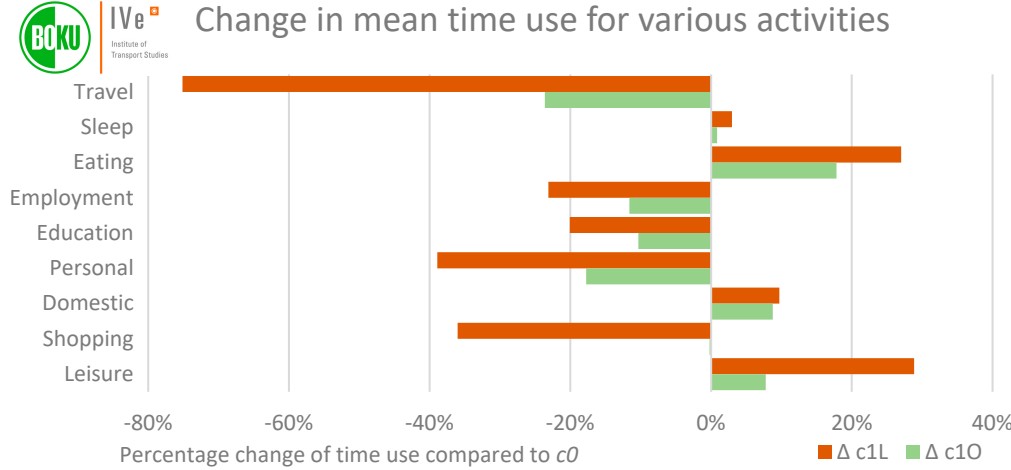

**Figure 8.** Change in mean time use compared to situation before the lockdown for various activities.

People also spent significantly less time on working, education, and personal activities during the lockdown and did not fully go back to the former situation in the opening phase. An exception is shopping: the reported time use after the lockdown returned to the before level. In exchange, people spent more time on sleeping, eating, domestic work, and leisure activities during the lockdown. The increased time use for eating and domestic work continued after the lockdown ended, whereas the time spent on sleeping and leisure went almost back to their original amounts.

### 3.6. Affectedness of Different Groups by Lockdown Measures

In this section we present the moderating influence of socio-economic characteristics on the changes in time use and mobility patterns. The situation before the lockdown (c0) is compared with the situation in the lockdown phase (c1L; Table 1) and the situation in the opening phase (c1O; Table 2). The ratio of the means of the dependent variable shows the intensity and direction of change. A value < 1 indicates that the mean of the dependent variable decreased in the considered phase compared to its level in c0; vice

versa, a value > 1 indicates an increase of the dependent variable. The last three columns report the results from linear regression models in the form of

$$y = \beta_o + \beta_c c + \beta_z z + \beta_{cz} cz + \varepsilon \tag{1}$$

with y being the dependent variable; $\beta_0$ being the intercept term; $\beta_c$, $\beta_z$, and $\beta_{cz}$ being the regression coefficients; c being a dummy of the considered COVID-19 phase; z being the socio-demographic variable to be tested; and $\varepsilon$ being an error term. All involved variables were standardised to make the $\beta$ parameters independent from the scales of the original variables.

The moderating effect of the socio-economic variable is captured by $\beta_{cz}$. It represents the net effect of the interaction between COVID-19 phase and socio-economic variable on top of the main effects of both variables; the corresponding *t*-value is reported in the rightmost column of the tables below. A positive value of $\beta_{cz}$ means that higher values of the socio-demographic variable cause a more positive (or less negative) change of the dependent variable in the COVID-19 phase compared to the pre-lockdown phase and vice versa.

We tested a wide range of moderator variables for their effect on the dependent variables (see Table A5 for the complete list). For reasons of space, presented in Tables 1 and 2 are only those variables for which the absolute *t*-value for the interaction effect is greater than 1.96, which means that the interaction effect is significant at the level of $\alpha < 0.05$.

Table 1 shows that formerly frequent travellers—persons with a university degree, employees, urban inhabitants, and public transport season ticket holders—reduced their time spent on travel as well as trip frequencies more than formerly infrequent travellers, e.g., elderly persons. Therefore, the travel behaviour of the two groups became more similar during the lockdown. As can be seen in Table 2, this pattern remained in the opening phase, where frequent travellers still show a significantly larger reduction. A similar effect can be observed for most time uses, but some effects are already disappearing in the opening phase. These findings are discussed in more detail in the following section including a comparison with results from literature.

## 4. Discussion of Results

In this section, we discuss our results, relate them to the literature, and discuss the study's limitations.

Regarding the affectedness of different socio-economic groups, we observe in general that the lockdown led to a convergence, meaning that existing differences before the lockdown were partly reduced in the majority of cases. This can be seen in the results from the linear regression for time spent on travel as well as trip frequencies where frequent travellers in the past reduced their travel more than less frequent travellers in the lockdown. The (preliminary) end of our story is therefore that the groups became more similar to each other than they were before the lockdown, although differences remained.

Similarly, the lockdown resulted in a reduction of *gender* differences, although it was far from equalizing them. Women spent significantly more time on domestic work in all phases (positive value of 0.10 for $\beta_z$ at Domestic in Table 1), but they increased their time less than men in the lockdown (negative value of $-0.21$ for $\beta_{cz}$). In the opening phase, there was no longer a significant interaction of gender and the difference in time use compared to the before-situation; consequently, Gender female is not included in Table 2. Furthermore, women spent less time for paid work than men, but they reduced their time for employment less in the lockdown. The gender differences however reappeared in the opening phase, almost to their pre-COVID levels.

Two specific patterns were observed for persons from households with children under 16 years. First, they increased their time for eating significantly more than other persons in the lockdown ($\beta_{cz} = 0.17$), while there was previously no difference. This might indicate that during the lockdown families were at home together and could socialize while having their meal. Second, they previously spent more time for employed work ($\beta_z = 0.14$) but reduced

their work time more than others ($\beta_{cz} = -0.08$), which is again a kind of convergence. These are in line with findings which found that the presence of children and full-time employment are two variables that influence the limit and shape of a person's daily time use distributions [42], and echoed by other COVID-19 studies [43].

However, there are also some exceptions where differences increased during the lockdown. Persons who already hold a university degree lowered their education activities less ($\beta_{cz} = 0.08$) in c1O than those who had not graduated (yet), thereby increasing the gap between both groups that had not been significant in c0. This might be because online education was more easily accessible for this group. Another example is that persons with above-average income reduced their personal activities more than others ($\beta_{cz} = -0.10$) in c1O which increased the previously existing difference ($\beta_z = -0.08$).

Regarding sleep, an interesting result is that public transport season ticket holders, who were also seen to decrease their travel more than others ($\beta_{cz} = -0.10$), had a bigger increase in time spent on sleeping than others ($\beta_{cz} = 0.08$). Persons with a higher age, who generally spend more time sleeping ($\beta_z = 0.11$) and for leisure activities ($\beta_z = 0.14$), increased it less than others ($\beta_{cz} = -0.10$ and $-0.08$, respectively) in the lockdown. It could be that for older persons, the additional time for sleep and leisure yielded less benefits than for those who had less of both in *c0*. Comparable results from the literature are missing, but the issue of how people use unexpected time gains would reward further investigation.

Regarding the use of public transport, we also observed a sharp drop as many other studies did [7,13,15,18,20,25,44]. According to our 'convergence pattern', those who used public transport more before the lockdown reduced it more than others during the lockdown. These are season ticket owners ($\beta_z = 0.61$, $\beta_{cz} = -0.32$), urban inhabitants ($\beta_z = 0.36$, $\beta_{cz} = -0.19$), and persons with a university degree ($\beta_z = 0.15$, $\beta_{cz} = -0.13$). This pattern remained in the opening phase, where the use of public transport remained at a lower level. It is worth noting that infrequent public transport users, who obviously have alternative options available (e.g., a private car) kept a larger share of their public transport trips in the lockdown ($\beta_z = -0.38$, $\beta_{cz} = 0.16$). There is evidence from several cities within the EU indicating that public transport ridership continued to be lower in 2021 compared to pre-pandemic levels while other modes recovered more quickly [45]. A prolonged modal shift away from public transport would be detrimental for a transition to a more sustainable mobility system.

A reduction of the overall travel demand in terms of trip frequency and trip distance is also a common finding of most COVID-19 related studies. There are indications that people from neighbourhoods with lower income, lower education, and more unemployment continued to travel while others reduced their trips [18]. Our data show that only university graduates reduced their travel demand in the lockdown significantly more than others ($\beta_{cz} = -0.11$). A reason for this might be that the other study only looked at public transport trips in a city, whereas we used a sample for all of Austria and for all modes. The finding that university education (and a corresponding type of job) is helpful for reducing travel demand was also reported by [25]. In another study, a slightly significant ($p < 0.1$) increase of teleworking with income was found, but no effect on trip frequency, meaning that even though commuting trips of higher income participants may have been reduced by telework, this is not necessarily true for the travel demand in general [16].

Regarding the location of the activities, the largest shifts towards activity performance at home could be observed for employed work and education. The at-home share also increased for eating and leisure, continuing into the opening phase. In a previous study it was reported that during the first lockdown in the UK, people were engaging in less risky activity/location/co-presence/duration combinations, meaning that they chose to perform activities with a high risk of infection increasingly at home, with no other persons or only household members present [23]. The focus of our analysis was different, but our data confirm this trend. Related to the location (at home vs. out of home) is the type of activity participation in terms of physical vs. virtual. For some activities such as education and leisure a shift to the home does not necessarily imply a shift in their type,

but if shopping or employment activities are performed at home they are in most cases performed virtually. From this we conclude that the lockdown unleashed a large potential of virtual home-based employment, which partly remained in the opening phase and could become a factor that reduces commuter travel demand in the future. This is important, because reducing motorised trips through remote working can avoid significant greenhouse gas emissions [46]. As to shopping activities, a study for the Chicago metropolitan area reports that while expenditure on groceries and meals (and their respective share of online purchasing) increased during the pandemic, the expenditures on other goods decreased [41]. This pattern remains inconclusive in our data, since the amount of time used for shopping is not a good indicator of the amount of purchased goods. The total amount of shopping activities is so small that even large relative changes are hidden in the background variation.

Still, we showed how a large share of out-of-home activities was shifted into the home, thereby rendering travel for these activities unnecessary. While there may be some loss of quality of the activity involved if it is performed at home, this demonstrates the potential for travel reduction through virtual activities that could lead to more sustainable lifestyles. However, a full analysis of impacts, e.g., on energy consumption and greenhouse gas emissions, of all involved activities would have to be carried out.

Regarding the engagement in secondary activities, we observed an increase in shares for both the lockdown and the opening phase. This is a plausible side-effect of the shift of activities to the home, where the conditions for multitasking are better than elsewhere. There are increases both in the share of activities performed at home and the multitasking share for the activities employed work, eating, and leisure, which supports this idea. Yet, in c0 we observed more secondary activities for out-of-home activities which challenges this interpretation. For the UK an increase of multitasking/secondary activities was observed in the first lockdown, but not in the third [22]. In our data, the higher share of secondary activities that occurred in the first lockdown in Austria persisted into the subsequent opening phase. A reason for the difference could be that our sample from the 'opening phase' was collected immediately after the first lockdown.

Regarding the changes in time use, we observe a reduction of employment, education and personal activities both in the lockdown and opening phase. Shopping was only reduced in the lockdown and bounced back to the before level in the opening phase. The time released from these activities as well as from the reduced mobility went towards sleep, eating, domestic, and leisure. This is consistent with time use studies that report a shift from paid work towards housework (especially in households with small children) as well as leisure activities and sleep [22,23]. In the opening phase, time use for sleep and leisure was reduced again, while eating and domestic work remained at high levels. This is comparable to the findings in [23] that time use changes were reversed in the opening phase following a full lockdown in the UK, but were not fully reset.

The limitations of our study relate to the way in which the data were collected. For one, we have only one measurement point for the 'opening phase' immediately after the first lockdown in summer 2020. After this, several more lockdowns were imposed in Austria. We saw that some time use changes were slow to move back to pre-COVID levels. We thus cannot answer the question of if and when the before-levels will be reached, or which changes due to COVID-19 will stay permanently. Secondly, the activities are reported as aggregate categories in the mobility activity diary, which is not comparable to a regular time use survey according to the HETUS guideline, in which the activities are reported in open text fields and subsequently coded. Activity categories such as personal, domestic, and leisure comprise many different sub-types for which diverging changes were found in some studies. Lastly, we did not ask explicitly for the type of activity in terms of virtual vs. physical performance. For some activities we can assume from the location that they have been performed virtually, but this does not apply to all activities. In the conclusions we will discuss how future research can improve on that.

### 5. Conclusions

This paper aims to contribute to a better understanding of mobility behaviour through analysing the effects on time use and mobility patterns that were caused by the first COVID-19 lockdown in Austria including the subsequent opening phase. Our findings were obtained from a comprehensive survey that combines data collection approaches from both mobility and time use research for an integrated perspective. We show that the lockdown effectively reduced travel in terms of trip rates and distances as well as public transport use, as well as out-of-home activity participation. During the subsequent opening phase, these effects partially persisted. Considering the shift from physical to virtual activities that could render the need to travel for these activities unnecessary and reduce emissions, our findings reveal some potential for specific groups such as employees with a university degree who are less likely to carry out manual labour. It remains to be seen to what extent the willingness to perform activities virtually will persist in a post-pandemic setting without restrictions.

We found that the time saved on travel, paid work, and personal care was initially shifted to sleep and leisure in the lockdown, but more to domestic activities and eating in the opening phase. The fact that changes were partially kept in the opening phase might indicate that people have habituated to their new patterns and are content with them. The changes we observed led in most cases to a convergence of socio-demographic groups in the sense that existing group differences diminished. However, the differences partly reappeared in the opening phase to the extent that people went back to their previous behaviour. Fundamental social inequalities such as the gender inequality will not disappear by shifting to virtual activities and need to be addressed in a different way.

Our conclusions for the transition to a more sustainable transport system firstly relate to the saving of emissions by shifting activities to virtual performance. Our data reveal a considerable potential which could be tapped into through creating the digital infrastructure, tax system, and legal requirements, as well as work environments that facilitate such a shift. One caveat with respect to the hypothesis of 'constant personal travel time budgets', i.e., that on average observed travel time expenditure changed little over time, may be that saved time from commuting or shopping trips might be used for additional trips elsewhere [47]. This will only result in less emissions if for the additional trips more sustainable modes or less distant destinations are chosen. Another condition for a positive environmental outcome is that the activities at home do not generate additional emissions, e.g., from more heating, that exceed the savings from travel.

Regarding the social dimension of sustainability, we saw a more equitable division of domestic work and paid work between males and females in the lockdown, but this trend was partly reset in the opening phase, so that there is further need of dedicated policies to enhance gender equality.

The observed persistence of some behavioural changes during the opening phase indicates a potential for permanent behaviour change. Although in this case the behaviour change was initially forced by a disruptive event, it also reveals some potential for the potentially significant behaviour changes needed to achieve a carbon neutral and sustainable transport system.

However, our research is limited insofar as the survey period only covers the first lockdown and opening phase, so we cannot evaluate long-term changes. Furthermore, we asked about activities only in broad categories which could obscure some diversity, and we did not explicitly ask if an activity was performed virtually.

Options for future research that build upon our findings should investigate how long the observed changes lasted and whether subsequent lockdowns had similar effects. Future surveys should furthermore ask explicitly if activities were performed virtually, i.e., using an electronic communication device. This would further the understanding of the prevalence and potential for shifting activities to the virtual realm. Lastly, the effect of secondary activities could be explored in more detail, as secondary activities can generate

additional utility while performing a less demanding task such as travelling. However, there are indications that excessive multitasking may reduce productivity and well-being.

**Author Contributions:** Conceptualization, A.G., R.H., Y.O.S. and L.H.; methodology, R.H. and Y.O.S.; software, R.H.; validation, R.H. and A.G.; formal analysis, L.H.; investigation, L.H. and A.G.; resources, A.G. and Y.O.S.; data curation, R.H. and L.H.; writing—original draft preparation, L.H.; writing—review and editing, A.G., R.H. and Y.O.S.; visualization, L.H.; supervision, A.G., R.H. and Y.O.S.; project administration, R.H.; funding acquisition, R.H., A.G. and Y.O.S. All authors have read and agreed to the published version of the manuscript.

**Funding:** This research was co-funded through financial contributions from the Austrian Federal Ministry for Social Affairs, Health, Care and Consumer Protection; the Vienna University of Economics and Business (internal order number 11000569); and as part of the project Digitalisation and Automation in the Transport and Mobility System (DAVeMoS; grant number 862678) of the University of Natural Resources and Life Sciences, Vienna. Any findings, conclusions, recommendations, or opinions expressed are those of the authors only.

**Institutional Review Board Statement:** Ethical review and approval were waived for this study because the data collection began before the establishment of an ethical review board at the University of Natural Resources and Life Sciences, Vienna. Still, the study was conducted according to the guidelines for good scientific practice of the Austrian Agency for Research Integrity.

**Informed Consent Statement:** Informed consent was obtained from all subjects involved in the study.

**Data Availability Statement:** The mobility-activity dataset used in this study can be requested for research purposes from the authors in exchange for a non-disclosure agreement.

**Acknowledgments:** We would like to acknowledge the outstanding work of further colleagues who were involved in the realisation of the survey, especially Gregor Husner, Martin Hinteregger, Michael Skok, and Florian Aschauer, as well as a diligent and skilled team of interviewers who were indispensable for the data collection.

**Conflicts of Interest:** The authors declare no conflict of interest.

## Appendix A

**Table A1.** Survey questions and possible answers (translated from German). Categories: PS: Personal socio-demographics; HS: Household socio-demographics; ME: mobility tool endowment; TC: trip characteristics; AC: activity characteristics.

| Category | Question | Possible Answers |
|---|---|---|
| PS | Year of birth | Year: |
| | Gender | Male |
| | | Female |
| | Height | in cm: |
| | Weight: | in kg: |
| | Highest completed education | No qualification (yet) |
| | | Secondary school |
| | | Apprenticeship |
| | | High school |
| | | University |
| | Type of employment | Student/apprentice |
| | | Employed |
| | | Self-employed |
| | | Not employed |
| | Regular working time before COVID-19 | h/week: |
| | Actual working time before COVID-19 | h/week: |
| | Regular working time during COVID-19 | h/week: |
| | Actual working time during COVID-19 | h/week: |
| | Net income from employment before COVID-19 | €/month: |
| | Net income from employment before COVID-19 | €/month: |

**Table A1.** *Cont.*

| Category | Question | Possible Answers |
|---|---|---|
| | . . . from letting | |
| | . . . from capital investments | |
| | . . . from child benefits | |
| | . . . from student benefits | |
| | . . . from pension | |
| | . . . from unemployment benefits | |
| HS | How many persons, including yourself, are living permanently in your household? | Total number: |
| | | . . . of which under 6 years: |
| | | . . . of which between 6 and 16 years: |
| | | . . . of which 16 years and older: |
| | Address of the household | |
| ME | How long does it take to walk from your dwelling to the following stops? | Bus: |
| | | Tram: |
| | | Subway: |
| | | Commuter/regional/long-distance train: |
| | How many of the following vehicles do you have in your household? | Cars: |
| | | Mopeds/motorbikes: |
| | | Bicycles: |
| | | . . . of which e-bikes: |
| | Which cars are present in your household? | Make: |
| | | Model: |
| | | Power [kW]: |
| | Fuel type | Petrol |
| | | Diesel |
| | | Electric |
| | | Hybrid |
| | If privately-owned car: | Bought new |
| | | Bought used |
| | | Received as gift |
| | If company-owned car: | Private use permitted |
| | | Private use not permitted |
| | Driver's licence | Car |
| | | Motorbike |
| | Bicycle availability | No |
| | | Yes |
| | Car availability | Always |
| | | Sometimes |
| | | Never |
| | Carsharing/rental car | No |
| | | Yes |
| | Motorbike/Scooter | No |
| | | Yes |
| | Availability of private parking at dwelling | No |
| | | Yes |
| | Availability of company parking at workplace | No |
| | | Yes |
| | Discount card for train travel owned? | No |
| | | Yes |
| | Public transport season ticket owned? | No |
| | | Yes |
| TC | Trip start time | Date: |
| | | Time: |
| | Trip end time | Date: |
| | | Time: |
| | Trip origin address | Street: |

**Table A1.** *Cont.*

| Category | Question | Possible Answers |
|---|---|---|
| | Trip destination address | No: |
| | | Locality: |
| | | Street: |
| | | No: |
| | | Locality: |
| | Used modes of transport | Walking |
| | | Kick scooter |
| | | Bicycle |
| | | E-Bike |
| | | Motorbike/moped |
| | | Car as driver |
| | | Car as passenger |
| | | Carsharing |
| | | Public Transport |
| | | Other (taxi, ride-hailing, airplane, . . . ) |
| | Used electronic devices | Route planning or navigation device |
| | | Online ticketing or booking |
| | Who else was present during the trip? | No one else |
| | | Partner |
| | | Children under 10 years |
| | | Other household member |
| | | Other acquaintance |
| | What else did you do during the trip (secondary activities)? | |
| AC | Activity at destination start time | Time: |
| | Activity at destination end time | Time: |
| | Activity category | Sleep |
| | | Eating |
| | | Employed work |
| | | Education |
| | | Personal |
| | | Domestic |
| | | Shopping |
| | | Leisure |
| | | Other |
| | How high was the level of physical activity during this activity? | Low |
| | | Medium |
| | | High |
| | What else did you do during this activity (secondary activities)? | |

**Table A2.** Description of predetermined activity categories.

| Activity Category | Description |
|---|---|
| Sleep | All sleep, including naps during the day. |
| Eating | Consumption of food. Cooking of food falls into category *domestic*. |
| Employment | Paid work, also when self-employed. Unpaid work falls into category *domestic* (care work, household work) or *leisure* (volunteering). |
| Education | Education in a school, university or apprenticeship. Personal self-improvement classes (e.g., language courses) fall into category *leisure*. |
| Personal | Personal activities that cannot be transferred to another person (e.g., personal hygiene, doctor's appointments, banking transactions). |
| Domestic | Work pertaining to the household (e.g., cooking, laundry, home repairs, washing the car). |
| Shopping | Buying of groceries and other goods, in-store as well as online. |
| Leisure | Activities, performed for pleasure (e.g., sports, meeting friends, church, watching TV) and voluntary help (e.g., helping neighbours, volunteer work). |

**Table A3.** Household characteristics of the MAED survey sample compared to the Austrian population (in percent).

| | MAED All Phases | c0 Only (*n* = 260) | c1L Only (*n* = 101) | c1O Only (*n* = 188) | Statistics Austria (STAT) | Difference (STAT to All Phases) |
|---|---|---|---|---|---|---|
| **Household size (persons)** | | | | | | |
| 1 | 30.6 | 28.8 | 32.7 | 31.9 | 37.8 | −7.2 |
| 2 | 36.6 | 37.3 | 41.6 | 33.0 | 30.4 | +6.2 |
| 3 | 15.7 | 18.1 | 8.9 | 16.0 | 14.6 | +1.1 |
| 4 | 14.9 | 12.7 | 14.9 | 18.1 | 11.3 | +3.7 |
| >4 | 2.2 | 3.1 | 2.0 | 1.1 | 6.0 | −3.8 |
| **Household members aged < 15** | | | | | | |
| 0 | 76.2 | 75.4 | 81.2 | 74.5 | 81.0 | −4.8 |
| 1 | 11.3 | 13.5 | 7.9 | 10.1 | 10.0 | +1.3 |
| 2 | 10.9 | 8.8 | 8.9 | 14.9 | 7.0 | +3.9 |
| >3 | 1.6 | 2.3 | 2.0 | 0.5 | 2.0 | −0.4 |
| **Federal state** | | | | | | |
| Burgenland | 4.2 | 4.2 | 5.0 | 3.7 | 3.2 | +1.0 |
| Lower Austria | 24.4 | 23.8 | 26.7 | 23.9 | 18.5 | +5.9 |
| Vienna | 20.9 | 23.1 | 14.9 | 21.3 | 23.0 | −2.1 |
| Carinthia | 6.2 | 5.0 | 9.9 | 5.9 | 6.4 | −0.2 |
| Styria | 13.7 | 13.8 | 15.8 | 12.2 | 13.9 | −0.3 |
| Upper Austria | 15.1 | 16.9 | 12.9 | 13.8 | 16.2 | −1.1 |
| Salzburg | 3.6 | 3.1 | 2.0 | 5.3 | 6.1 | −2.5 |
| Tyrol | 8.4 | 8.5 | 6.9 | 9.0 | 8.4 | ±0.0 |
| Vorarlberg | 3.5 | 1.5 | 5.9 | 4.8 | 4.3 | −0.8 |

**Table A4.** Personal characteristics of the MAED survey sample compared to the Austrian population (in percent).

| | MAED All Phases | c0 Only (*n* = 441) | c1L Only (*n* = 166) | c1O Only (*n* = 249) | Statistics Austria (STAT) | Difference (STAT to All Phases) |
|---|---|---|---|---|---|---|
| **Gender** | | | | | | |
| Male | 44.4 | 43.8 | 46.4 | 44.2 | 49.2 | −4.8 |
| Female | 55.6 | 56.2 | 53.6 | 55.8 | 50.8 | +4.8 |
| **Age** | | | | | | |
| 16–19 | 2.3 | 1.8 | 4.2 | 2.0 | 5.7 | −3.4 |
| 20–29 | 13.5 | 13.4 | 14.5 | 13.3 | 14.4 | −0.9 |
| 30–39 | 18.6 | 20.9 | 15.7 | 17.0 | 16.1 | +2.6 |
| 40–49 | 17.4 | 14.3 | 21.1 | 20.1 | 15.4 | +2.0 |
| 50–59 | 20.8 | 23.4 | 15.1 | 20.1 | 18.3 | +2.5 |
| 60+ | 27.3 | 26.3 | 29.5 | 27.6 | 30.1 | −2.8 |
| **Type of employment** | | | | | | |
| Employed | 53.3 | 52.8 | 55.4 | 52.7 | 43.1 | +10.1 |
| Self-employed | 6.9 | 6.8 | 5.4 | 7.8 | 6.0 | +0.9 |
| Not employed | 39.8 | 40.4 | 39.2 | 39.5 | 50.9 | −11.0 |
| **Highest degree of education** | | | | | | |
| Compulsory school | 8.5 | 9.1 | 8.4 | 7.8 | 17.6 | −9.1 |
| Apprenticeship | 42.1 | 42.4 | 40.4 | 42.5 | 49.9 | −7.9 |
| High school | 22.3 | 21.5 | 27.1 | 20.7 | 16.0 | +6.3 |
| University | 27.1 | 27.0 | 24.1 | 28.9 | 16.5 | +10.6 |
| **Level of urbanisation** | | | | | | |
| Urban | 27.5 | 29.3 | 19.3 | 29.6 | 31.4 | −3.9 |
| Intermediate | 31.3 | 32.2 | 38.6 | 25.9 | 30.8 | +0.5 |
| Rural | 41.2 | 38.5 | 42.2 | 44.6 | 37.8 | +3.4 |

Table A5. All tested moderator variables and *t*-values for the interaction effect.

| Dependent Variable (y) | Moderator Variable (z) | c0 vs. c1L | | | c0 vs. c1O | | |
|---|---|---|---|---|---|---|---|
| | | $\beta_z$ | $\beta_{cz}$ | t-val. ($\beta_{cz}$) | $\beta_z$ | $\beta_{cz}$ | t-val. ($\beta_{cz}$) |
| Domestic | Age | 0.304 | 0.036 | 0.94 | 0.294 | 0.015 | 0.43 |
| Domestic | No. of visited destinations per week | 0.040 | 0.039 | 0.97 | 0.006 | −0.020 | −0.55 |
| Domestic | University educated | −0.067 | 0.005 | 0.13 | −0.091 | −0.022 | −0.59 |
| Domestic | Employed | −0.265 | 0.036 | 0.92 | −0.238 | 0.063 | 1.77 |
| Domestic | Gender | 0.288 | −0.077 | −1.98 | 0.310 | −0.033 | −0.95 |
| Domestic | Household urbanity | 0.230 | −0.041 | −1.04 | 0.253 | −0.013 | −0.37 |
| Domestic | Income | −0.206 | 0.046 | 1.17 | −0.160 | 0.094 | 2.61 |
| Domestic | Current workplace flexibility | −0.217 | −0.009 | −0.23 | −0.166 | 0.050 | 1.39 |
| Domestic | Max workplace flexibility | −0.221 | 0.002 | 0.06 | −0.175 | 0.062 | 1.71 |
| Domestic | Physical activity at job | −0.245 | 0.018 | 0.45 | −0.193 | 0.078 | 2.17 |
| Domestic | Driver's license owned | 0.020 | 0.001 | 0.03 | 0.055 | 0.051 | 1.39 |
| Domestic | Car availability | 0.054 | 0.030 | 0.73 | 0.081 | 0.051 | 1.38 |
| Domestic | All persons in hh | 0.068 | −0.015 | −0.36 | 0.063 | −0.018 | −0.48 |
| Domestic | Persons under 16 in hh | 0.115 | 0.015 | 0.36 | 0.150 | 0.049 | 1.34 |
| Domestic | Person under 6 in hh | 0.186 | −0.017 | −0.44 | 0.206 | 0.010 | 0.27 |
| Domestic | Discount card train owned | −0.049 | 0.019 | 0.48 | −0.093 | −0.033 | −0.90 |
| Domestic | Discount card local PT owned | −0.156 | −0.009 | −0.22 | −0.211 | −0.061 | −1.70 |
| Domestic | Effective current working time | −0.378 | −0.013 | −0.35 | −0.311 | 0.099 | 2.86 |
| Domestic | Regular current working time | −0.379 | 0.007 | 0.20 | −0.314 | 0.115 | 3.32 |
| Eating | Age | 0.174 | −0.029 | −0.74 | 0.252 | 0.059 | 1.69 |
| Eating | No. of visited destinations per week | −0.068 | −0.025 | −0.62 | −0.020 | 0.046 | 1.27 |
| Eating | University educated | −0.013 | −0.015 | −0.38 | −0.003 | 0.002 | 0.05 |
| Eating | Employed | −0.196 | 0.000 | 0.00 | −0.207 | 0.002 | 0.07 |
| Eating | Gender | 0.030 | −0.025 | −0.64 | 0.054 | 0.007 | 0.20 |
| Eating | Household urbanity | 0.091 | 0.005 | 0.12 | 0.057 | −0.048 | −1.33 |
| Eating | Income | −0.024 | −0.028 | −0.72 | −0.013 | −0.007 | −0.20 |
| Eating | Current workplace flexibility | −0.097 | 0.028 | 0.71 | −0.122 | 0.002 | 0.06 |
| Eating | Max workplace flexibility | −0.141 | 0.011 | 0.27 | −0.161 | 0.003 | 0.08 |
| Eating | Physical activity at job | −0.132 | 0.013 | 0.32 | −0.133 | 0.015 | 0.43 |
| Eating | Driver's license owned | 0.083 | −0.021 | −0.53 | 0.092 | −0.008 | −0.22 |
| Eating | Car availability | 0.097 | −0.002 | −0.06 | 0.076 | −0.036 | −0.99 |
| Eating | All persons in hh | 0.021 | 0.066 | 1.67 | −0.028 | −0.008 | −0.21 |
| Eating | Persons under 16 in hh | 0.021 | 0.169 | 4.31 | −0.041 | 0.044 | 1.22 |
| Eating | Person under 6 in hh | 0.070 | 0.043 | 1.10 | 0.028 | −0.031 | −0.86 |
| Eating | Discount card train owned | 0.042 | 0.033 | 0.84 | 0.048 | 0.029 | 0.79 |
| Eating | Discount card local PT owned | −0.103 | −0.005 | −0.12 | −0.099 | 0.014 | 0.39 |
| Eating | Effective current working time | −0.229 | −0.016 | −0.42 | −0.210 | 0.046 | 1.31 |
| Eating | Regular current working time | −0.230 | 0.009 | 0.23 | −0.225 | 0.053 | 1.51 |
| Education | Age | −0.387 | 0.029 | 0.76 | −0.372 | 0.009 | 0.27 |
| Education | No. of visited destinations per week | −0.111 | −0.057 | −1.40 | −0.048 | 0.038 | 1.04 |
| Education | University educated | −0.011 | 0.008 | 0.19 | 0.049 | 0.075 | 2.05 |
| Education | Employed | −0.164 | 0.015 | 0.38 | −0.209 | −0.054 | −1.51 |
| Education | Gender | −0.025 | 0.003 | 0.08 | −0.032 | −0.009 | −0.23 |
| Education | Household urbanity | −0.141 | 0.006 | 0.14 | −0.147 | −0.007 | −0.18 |
| Education | Income | −0.306 | 0.017 | 0.44 | −0.295 | 0.001 | 0.03 |
| Education | Current workplace flexibility | −0.090 | 0.025 | 0.61 | −0.103 | −0.002 | −0.05 |
| Education | Max workplace flexibility | −0.119 | 0.020 | 0.49 | −0.093 | 0.039 | 1.06 |
| Education | Physical activity at job | −0.073 | −0.030 | −0.75 | −0.085 | −0.046 | −1.24 |
| Education | Driver's license owned | −0.160 | 0.009 | 0.22 | −0.143 | 0.007 | 0.19 |
| Education | Car availability | −0.215 | −0.020 | −0.51 | −0.156 | 0.045 | 1.25 |
| Education | All persons in hh | 0.049 | 0.048 | 1.19 | 0.011 | −0.008 | −0.23 |
| Education | Persons under 16 in hh | −0.033 | 0.054 | 1.33 | −0.065 | −0.006 | −0.17 |
| Education | Person under 6 in hh | −0.060 | −0.018 | −0.43 | −0.043 | 0.008 | 0.21 |
| Education | Discount card train owned | 0.090 | −0.021 | −0.51 | 0.086 | −0.016 | −0.42 |
| Education | Discount card local PT owned | 0.278 | 0.059 | 1.51 | 0.240 | −0.002 | −0.04 |
| Education | Effective current working time | −0.133 | 0.004 | 0.10 | −0.134 | 0.005 | 0.13 |

**Table A5.** *Cont.*

| Dependent Variable (y) | Moderator Variable (z) | c0 vs. c1L | | | c0 vs. c1O | | |
|---|---|---|---|---|---|---|---|
| | | $\beta_z$ | $\beta_{cz}$ | t-val. ($\beta_{cz}$) | $\beta_z$ | $\beta_{cz}$ | t-val. ($\beta_{cz}$) |
| Education | Regular current working time | −0.142 | −0.012 | −0.29 | −0.141 | 0.002 | 0.05 |
| Leisure | Age | 0.142 | −0.081 | −2.12 | 0.145 | −0.060 | −1.67 |
| Leisure | No. of visited destinations per week | −0.051 | −0.015 | −0.39 | −0.048 | −0.004 | −0.11 |
| Leisure | University educated | −0.156 | −0.017 | −0.43 | −0.097 | 0.064 | 1.77 |
| Leisure | Employed | −0.267 | 0.016 | 0.43 | −0.247 | 0.041 | 1.15 |
| Leisure | Gender | −0.064 | −0.003 | −0.07 | −0.078 | −0.018 | −0.50 |
| Leisure | Household urbanity | −0.017 | −0.057 | −1.47 | 0.007 | −0.010 | −0.28 |
| Leisure | Income | −0.092 | −0.021 | −0.56 | −0.025 | 0.069 | 1.89 |
| Leisure | Current workplace flexibility | −0.168 | 0.076 | 2.01 | −0.203 | 0.024 | 0.66 |
| Leisure | Max workplace flexibility | −0.191 | 0.065 | 1.70 | −0.195 | 0.053 | 1.48 |
| Leisure | Physical activity at job | −0.194 | −0.024 | −0.65 | −0.226 | −0.067 | −1.87 |
| Leisure | Driver's license owned | −0.022 | 0.017 | 0.44 | −0.054 | −0.030 | −0.83 |
| Leisure | Car availability | −0.027 | −0.060 | −1.56 | −0.004 | −0.017 | −0.46 |
| Leisure | All persons in hh | −0.128 | 0.006 | 0.17 | −0.103 | 0.035 | 0.97 |
| Leisure | Persons under 16 in hh | −0.165 | −0.001 | −0.03 | −0.180 | −0.012 | −0.34 |
| Leisure | Person under 6 in hh | −0.102 | 0.017 | 0.45 | −0.127 | −0.018 | −0.49 |
| Leisure | Discount card train owned | 0.010 | 0.063 | 1.64 | −0.014 | 0.021 | 0.56 |
| Leisure | Discount card local PT owned | −0.020 | 0.051 | 1.31 | −0.042 | 0.010 | 0.27 |
| Leisure | Effective current working time | −0.266 | −0.028 | −0.75 | −0.233 | 0.043 | 1.21 |
| Leisure | Regular current working time | −0.248 | 0.030 | 0.80 | −0.222 | 0.076 | 2.13 |
| Personal | Age | 0.114 | 0.016 | 0.41 | 0.137 | 0.041 | 1.14 |
| Personal | No. of visited destinations per week | 0.024 | −0.002 | −0.06 | 0.059 | 0.039 | 1.06 |
| Personal | University educated | 0.005 | 0.004 | 0.11 | −0.056 | −0.069 | −1.89 |
| Personal | Employed | −0.113 | −0.012 | −0.30 | −0.160 | −0.064 | −1.78 |
| Personal | Gender | −0.008 | 0.010 | 0.24 | 0.025 | 0.047 | 1.30 |
| Personal | Household urbanity | 0.028 | 0.019 | 0.48 | −0.009 | −0.032 | −0.86 |
| Personal | Income | −0.003 | 0.001 | 0.02 | −0.083 | −0.101 | −2.78 |
| Personal | Current workplace flexibility | −0.051 | −0.013 | −0.33 | −0.076 | −0.038 | −1.05 |
| Personal | Max workplace flexibility | −0.097 | −0.010 | −0.25 | −0.130 | −0.039 | −1.06 |
| Personal | Physical activity at job | −0.076 | 0.016 | 0.40 | −0.103 | −0.027 | −0.75 |
| Personal | Driver's license owned | −0.050 | 0.008 | 0.20 | −0.079 | −0.038 | −1.03 |
| Personal | Car availability | −0.041 | 0.009 | 0.21 | −0.067 | −0.022 | −0.61 |
| Personal | All persons in hh | −0.107 | −0.028 | −0.72 | −0.101 | −0.016 | −0.43 |
| Personal | Persons under 16 in hh | −0.060 | −0.031 | −0.79 | −0.073 | −0.035 | −0.95 |
| Personal | Person under 6 in hh | −0.044 | −0.040 | −1.01 | −0.062 | −0.051 | −1.39 |
| Personal | Discount card train owned | 0.003 | −0.038 | −0.95 | 0.020 | −0.009 | −0.26 |
| Personal | Discount card local PT owned | −0.074 | 0.001 | 0.03 | −0.035 | 0.050 | 1.38 |
| Personal | Effective current working time | −0.131 | 0.001 | 0.03 | −0.175 | −0.034 | −0.95 |
| Personal | Regular current working time | −0.136 | 0.002 | 0.06 | −0.167 | −0.012 | −0.35 |
| Shopping | Age | 0.155 | −0.048 | −1.20 | 0.173 | −0.007 | −0.18 |
| Shopping | No. of visited destinations per week | 0.283 | 0.025 | 0.64 | 0.252 | −0.011 | −0.31 |
| Shopping | University educated | 0.025 | 0.006 | 0.16 | −0.009 | −0.036 | −0.98 |
| Shopping | Employed | −0.151 | 0.049 | 1.23 | −0.177 | −0.001 | −0.03 |
| Shopping | Gender | 0.151 | −0.101 | −2.57 | 0.196 | −0.015 | −0.41 |
| Shopping | Household urbanity | −0.018 | 0.039 | 0.97 | −0.046 | −0.008 | −0.21 |
| Shopping | Income | −0.043 | 0.057 | 1.42 | −0.055 | 0.025 | 0.67 |
| Shopping | Current workplace flexibility | −0.108 | 0.057 | 1.44 | −0.104 | 0.044 | 1.20 |
| Shopping | Max workplace flexibility | −0.097 | 0.041 | 1.04 | −0.116 | 0.010 | 0.26 |
| Shopping | Physical activity at job | −0.111 | 0.004 | 0.11 | −0.093 | 0.020 | 0.53 |
| Shopping | Driver's license owned | 0.057 | −0.002 | −0.04 | −0.001 | −0.074 | −2.02 |
| Shopping | Car availability | 0.025 | 0.040 | 1.01 | −0.031 | −0.037 | −1.01 |
| Shopping | All persons in hh | −0.180 | 0.010 | 0.25 | −0.171 | 0.010 | 0.29 |
| Shopping | Persons under 16 in hh | −0.113 | 0.047 | 1.17 | −0.120 | 0.023 | 0.63 |
| Shopping | Person under 6 in hh | −0.095 | 0.044 | 1.10 | −0.086 | 0.040 | 1.09 |
| Shopping | Discount card train owned | 0.027 | −0.073 | −1.83 | 0.060 | −0.016 | −0.42 |
| Shopping | Discount card local PT owned | −0.037 | −0.047 | −1.17 | −0.019 | −0.009 | −0.25 |

| Dependent Variable (y) | Moderator Variable (z) | c0 vs. c1L | | | c0 vs. c1O | | |
|---|---|---|---|---|---|---|---|
| | | $\beta_z$ | $\beta_{cz}$ | t-val. ($\beta_{cz}$) | $\beta_z$ | $\beta_{cz}$ | t-val. ($\beta_{cz}$) |
| Shopping | Effective current working time | −0.207 | 0.060 | 1.53 | −0.209 | 0.048 | 1.33 |
| Shopping | Regular current working time | −0.196 | 0.076 | 1.95 | −0.212 | 0.048 | 1.35 |
| Sleeping | Age | 0.105 | −0.099 | −2.46 | 0.086 | −0.098 | −2.68 |
| Sleeping | No. of visited destinations per week | −0.200 | −0.036 | −0.89 | −0.164 | 0.023 | 0.62 |
| Sleeping | University educated | −0.078 | −0.033 | −0.81 | −0.059 | 0.002 | 0.05 |
| Sleeping | Employed | −0.350 | 0.042 | 1.11 | −0.310 | 0.082 | 2.36 |
| Sleeping | Gender | 0.130 | −0.062 | −1.54 | 0.097 | −0.087 | −2.39 |
| Sleeping | Household urbanity | 0.010 | −0.035 | −0.87 | 0.009 | −0.026 | −0.70 |
| Sleeping | Income | −0.227 | −0.012 | −0.30 | −0.213 | 0.006 | 0.17 |
| Sleeping | Current workplace flexibility | −0.183 | 0.034 | 0.86 | −0.197 | 0.010 | 0.29 |
| Sleeping | Max workplace flexibility | −0.250 | 0.046 | 1.18 | −0.242 | 0.052 | 1.47 |
| Sleeping | Physical activity at job | −0.207 | 0.043 | 1.10 | −0.166 | 0.083 | 2.30 |
| Sleeping | Driver's license owned | −0.198 | 0.072 | 1.82 | −0.171 | 0.082 | 2.28 |
| Sleeping | Car availability | −0.178 | 0.061 | 1.53 | −0.145 | 0.088 | 2.43 |
| Sleeping | All persons in hh | −0.058 | 0.017 | 0.41 | −0.009 | 0.073 | 1.98 |
| Sleeping | Persons under 16 in hh | −0.080 | 0.022 | 0.54 | −0.112 | −0.022 | −0.61 |
| Sleeping | Person under 6 in hh | −0.061 | −0.023 | −0.56 | −0.044 | 0.008 | 0.20 |
| Sleeping | Discount card train owned | 0.046 | 0.043 | 1.06 | 0.050 | 0.035 | 0.96 |
| Sleeping | Discount card local PT owned | 0.013 | 0.080 | 1.98 | 0.008 | 0.048 | 1.30 |
| Sleeping | Effective current working time | −0.325 | 0.076 | 2.00 | −0.308 | 0.097 | 2.79 |
| Sleeping | Regular current working time | −0.322 | 0.094 | 2.47 | −0.296 | 0.124 | 3.56 |
| Travel | Age | −0.188 | 0.117 | 3.59 | −0.214 | 0.074 | 2.10 |
| Travel | No. of visited destinations per week | 0.483 | −0.008 | −0.28 | 0.528 | 0.024 | 0.78 |
| Travel | University educated | 0.181 | −0.126 | −3.87 | 0.212 | −0.076 | −2.17 |
| Travel | Employed | 0.225 | −0.072 | −2.23 | 0.215 | −0.085 | −2.43 |
| Travel | Gender | −0.069 | −0.007 | −0.22 | −0.039 | 0.035 | 0.98 |
| Travel | Household urbanity | −0.260 | 0.121 | 3.83 | −0.298 | 0.071 | 2.06 |
| Travel | Income | 0.150 | −0.038 | −1.13 | 0.128 | −0.067 | −1.87 |
| Travel | Current workplace flexibility | 0.116 | −0.077 | −2.32 | 0.132 | −0.053 | −1.49 |
| Travel | Max workplace flexibility | 0.166 | −0.097 | −2.95 | 0.162 | −0.096 | −2.73 |
| Travel | Physical activity at job | 0.144 | 0.001 | 0.04 | 0.128 | −0.026 | −0.73 |
| Travel | Driver's license owned | 0.039 | 0.009 | 0.26 | 0.003 | −0.042 | −1.15 |
| Travel | Car availability | −0.014 | 0.048 | 1.43 | −0.086 | −0.044 | −1.24 |
| Travel | All persons in hh | 0.008 | −0.033 | −0.97 | −0.021 | −0.062 | −1.72 |
| Travel | Persons under 16 in hh | 0.016 | −0.002 | −0.06 | −0.001 | −0.023 | −0.65 |
| Travel | Person under 6 in hh | 0.007 | 0.013 | 0.38 | 0.006 | 0.008 | 0.21 |
| Travel | Discount card train owned | 0.105 | −0.054 | −1.62 | 0.138 | −0.015 | −0.42 |
| Travel | Discount card local PT owned | 0.231 | −0.139 | −4.35 | 0.267 | −0.084 | −2.44 |
| Travel | Effective current working time | 0.196 | −0.045 | −1.37 | 0.152 | −0.107 | −3.02 |
| Travel | Regular current working time | 0.215 | −0.062 | −1.90 | 0.173 | −0.121 | −3.42 |
| Employment | Age | −0.285 | 0.044 | 1.13 | −0.304 | 0.019 | 0.55 |
| Employment | No. of visited destinations per week | −0.002 | 0.034 | 0.84 | −0.057 | −0.047 | −1.27 |
| Employment | University educated | 0.175 | 0.056 | 1.40 | 0.128 | −0.025 | −0.69 |
| Employment | Employed | 0.729 | −0.045 | −1.65 | 0.753 | −0.031 | −1.28 |
| Employment | Gender | −0.214 | 0.101 | 2.57 | −0.236 | 0.058 | 1.64 |
| Employment | Household urbanity | −0.069 | 0.046 | 1.14 | −0.061 | 0.046 | 1.25 |
| Employment | Income | 0.456 | −0.007 | −0.21 | 0.412 | −0.071 | −2.14 |
| Employment | Current workplace flexibility | 0.463 | −0.069 | −1.92 | 0.498 | −0.029 | −0.90 |
| Employment | Max workplace flexibility | 0.551 | −0.061 | −1.81 | 0.545 | −0.089 | −2.90 |
| Employment | Physical activity at job | 0.525 | −0.006 | −0.19 | 0.533 | 0.005 | 0.17 |
| Employment | Driver's license owned | 0.149 | −0.050 | −1.25 | 0.165 | −0.015 | −0.40 |
| Employment | Car availability | 0.155 | −0.008 | −0.19 | 0.121 | −0.054 | −1.49 |
| Employment | All persons in hh | 0.121 | −0.022 | −0.55 | 0.125 | −0.015 | −0.41 |
| Employment | Persons under 16 in hh | 0.135 | −0.083 | −2.09 | 0.187 | −0.004 | −0.10 |
| Employment | Person under 6 in hh | 0.010 | 0.016 | 0.39 | 0.023 | 0.028 | 0.77 |

**Table A5.** *Cont.*

| Dependent Variable (y) | Moderator Variable (z) | c0 vs. c1L | | | c0 vs. c1O | | |
|---|---|---|---|---|---|---|---|
| | | $\beta_z$ | $\beta_{cz}$ | t-val. ($\beta_{cz}$) | $\beta_z$ | $\beta_{cz}$ | t-val. ($\beta_{cz}$) |
| Employment | Discount card train owned | −0.077 | −0.046 | −1.15 | −0.044 | 0.006 | 0.17 |
| Employment | Discount card local PT owned | 0.005 | −0.058 | −1.43 | 0.053 | 0.015 | 0.41 |
| Employment | Effective current working time | 0.821 | 0.010 | 0.45 | 0.785 | −0.128 | −5.76 |
| Employment | Regular current working time | 0.807 | −0.059 | −2.53 | 0.771 | −0.188 | −8.44 |
| DistSumWk (sum of travel distance per week) | Age | −0.073 | 0.036 | 0.91 | −0.078 | 0.003 | 0.10 |
| DistSumWk | No. of visited destinations per week | 0.179 | −0.006 | −0.15 | 0.179 | 0.029 | 0.82 |
| DistSumWk | University educated | 0.039 | −0.033 | −0.85 | 0.092 | 0.048 | 1.31 |
| DistSumWk | Employed | 0.161 | −0.068 | −1.76 | 0.145 | −0.034 | −0.93 |
| DistSumWk | Gender | −0.047 | −0.003 | −0.08 | −0.052 | −0.016 | −0.45 |
| DistSumWk | Household urbanity | 0.004 | 0.010 | 0.27 | 0.041 | 0.050 | 1.36 |
| DistSumWk | Income | 0.151 | −0.060 | −1.54 | 0.143 | −0.018 | −0.49 |
| DistSumWk | Current workplace flexibility | 0.084 | −0.062 | −1.60 | 0.093 | −0.017 | −0.48 |
| DistSumWk | Max workplace flexibility | 0.092 | −0.055 | −1.42 | 0.108 | −0.006 | −0.17 |
| DistSumWk | Physical activity at job | 0.083 | −0.010 | −0.24 | 0.052 | −0.029 | −0.80 |
| DistSumWk | Driver's license owned | 0.107 | −0.043 | −1.09 | 0.096 | −0.016 | −0.45 |
| DistSumWk | Car availability | 0.116 | −0.038 | −0.98 | 0.109 | −0.014 | −0.39 |
| DistSumWk | All persons in hh | 0.020 | −0.013 | −0.33 | −0.034 | −0.071 | −1.95 |
| DistSumWk | Persons under 16 in hh | 0.017 | 0.005 | 0.13 | −0.011 | −0.028 | −0.77 |
| DistSumWk | Person under 6 in hh | 0.031 | 0.004 | 0.09 | 0.015 | −0.014 | −0.37 |
| DistSumWk | Discount card train owned | 0.002 | −0.002 | −0.06 | 0.066 | 0.071 | 1.93 |
| DistSumWk | Discount card local PT owned | −0.082 | 0.028 | 0.72 | −0.030 | 0.066 | 1.81 |
| DistSumWk | Effective current working time | 0.157 | −0.048 | −1.25 | 0.127 | −0.045 | −1.25 |
| DistSumWk | Regular current working time | 0.168 | −0.062 | −1.62 | 0.134 | −0.058 | −1.61 |
| Mode Share PT | Age | −0.222 | 0.121 | 3.15 | −0.227 | 0.077 | 2.15 |
| Mode Share PT | No. of visited destinations per week | 0.066 | −0.044 | −1.11 | 0.045 | −0.041 | −1.13 |
| Mode Share PT | University educated | 0.152 | −0.126 | −3.25 | 0.123 | −0.123 | −3.40 |
| Mode Share PT | Employed | 0.016 | −0.003 | −0.06 | 0.018 | 0.000 | 0.01 |
| Mode Share PT | Gender | 0.043 | −0.025 | −0.64 | 0.015 | −0.051 | −1.39 |
| Mode Share PT | Household urbanity | −0.361 | 0.187 | 5.17 | −0.406 | 0.078 | 2.34 |
| Mode Share PT | Income | −0.059 | 0.011 | 0.28 | −0.072 | −0.010 | −0.27 |
| Mode Share PT | Current workplace flexibility | 0.031 | −0.036 | −0.91 | −0.010 | −0.072 | −1.98 |
| Mode Share PT | Max workplace flexibility | 0.068 | −0.068 | −1.72 | 0.029 | −0.091 | −2.50 |
| Mode Share PT | Physical activity at job | 0.029 | 0.019 | 0.48 | 0.006 | −0.015 | −0.40 |
| Mode Share PT | Driver's license owned | −0.243 | 0.065 | 1.70 | −0.280 | −0.023 | −0.64 |
| Mode Share PT | Car availability | −0.379 | 0.163 | 4.54 | −0.464 | 0.020 | 0.60 |
| Mode Share PT | All persons in hh | −0.046 | 0.011 | 0.28 | −0.061 | −0.012 | −0.32 |
| Mode Share PT | Persons under 16 in hh | −0.058 | 0.001 | 0.02 | −0.088 | −0.036 | −0.97 |
| Mode Share PT | Person under 6 in hh | −0.058 | 0.010 | 0.25 | −0.066 | −0.005 | −0.14 |
| Mode Share PT | Discount card train owned | 0.097 | −0.052 | −1.32 | 0.105 | −0.029 | −0.80 |
| Mode Share PT | Discount card local PT owned | 0.610 | −0.315 | −11.16 | 0.666 | −0.158 | −5.93 |
| Mode Share PT | Effective current working time | 0.013 | 0.011 | 0.27 | 0.031 | 0.024 | 0.66 |
| Mode Share PT | Regular current working time | 0.028 | −0.005 | −0.14 | 0.048 | 0.014 | 0.37 |
| Trip frequency (per week) | Age | −0.113 | 0.078 | 2.41 | −0.141 | 0.040 | 1.13 |
| Trip frequency | No. of visited destinations per week | 0.757 | 0.021 | 1.04 | 0.803 | 0.004 | 0.21 |
| Trip frequency | University educated | 0.156 | −0.107 | −3.35 | 0.189 | −0.063 | −1.80 |
| Trip frequency | Employed | 0.218 | −0.033 | −1.05 | 0.229 | −0.036 | −1.03 |
| Trip frequency | Gender | 0.019 | −0.029 | −0.88 | 0.059 | 0.023 | 0.66 |
| Trip frequency | Household urbanity | −0.149 | 0.062 | 1.90 | −0.179 | 0.032 | 0.91 |
| Trip frequency | Income | 0.157 | −0.020 | −0.62 | 0.159 | −0.028 | −0.80 |

**Table A5.** *Cont.*

| Dependent Variable (y) | Moderator Variable (z) | c0 vs. c1L | | | c0 vs. c1O | | |
|---|---|---|---|---|---|---|---|
| | | $\beta_z$ | $\beta_{cz}$ | t-val. ($\beta_{cz}$) | $\beta_z$ | $\beta_{cz}$ | t-val. ($\beta_{cz}$) |
| Trip frequency | Current workplace flexibility | 0.104 | −0.074 | −2.28 | 0.163 | −0.007 | −0.21 |
| Trip frequency | Max workplace flexibility | 0.135 | −0.082 | −2.54 | 0.176 | −0.038 | −1.08 |
| Trip frequency | Physical activity at job | 0.123 | 0.033 | 1.01 | 0.134 | 0.034 | 0.95 |
| Trip frequency | Driver's license owned | 0.108 | −0.016 | −0.49 | 0.100 | −0.028 | −0.79 |
| Trip frequency | Car availability | 0.092 | 0.019 | 0.59 | 0.060 | −0.034 | −0.94 |
| Trip frequency | All persons in hh | 0.048 | −0.055 | −1.67 | 0.062 | −0.033 | −0.92 |
| Trip frequency | Persons under 16 in hh | 0.100 | −0.059 | −1.80 | 0.149 | 0.003 | 0.09 |
| Trip frequency | Person under 6 in hh | 0.009 | 0.006 | 0.19 | 0.048 | 0.055 | 1.54 |
| Trip frequency | Discount card train owned | 0.120 | −0.096 | −2.96 | 0.146 | −0.062 | −1.76 |
| Trip frequency | Discount card local PT owned | 0.080 | −0.097 | −2.97 | 0.101 | −0.058 | −1.64 |
| Trip frequency | Effective current working time | 0.165 | −0.030 | −0.94 | 0.139 | −0.079 | −2.23 |
| Trip frequency | Regular current working time | 0.175 | −0.029 | −0.89 | 0.148 | −0.081 | −2.30 |

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
