# Peer review of "The Impacts of a COVID-19 Related Lockdown (and Reopening Phases) on Time Use and Mobility for Activities in Austria—Results from a Multi-Wave Combined Survey"

_sustainability, doi:10.3390/su14127422_

Round 1

Reviewer 1 Report

The manuscript analyses the changes in mobility during and after the lockdown in Austria. The topic is interesting. Overall, the manuscript is good-written. However, I have some comments and suggestions:

The authors overviewed the previous literature and presented the main insights in Introduction section. However more detailed analysis of previous studies should be provided, please consider these sources also:

1.     Gurbuz, H; Sohret, Y; Ekici, S. Evaluating effects of the Covid-19 pandemic period on energy consumption and enviro-economic indicators of Turkish road transportation. ENERGY SOURCES PART A-RECOVERY UTILIZATION AND ENVIRONMENTAL EFFECTS, 2021, DOI: 10.1080/15567036.2021.1889077

2.     Styring, P; Duckworth, EL; Platt, EG. Synthetic Fuels in a Transport Transition: Fuels to Prevent a Transport Underclass. FRONTIERS IN ENERGY RESEARCH, 2021, 9, 707867.

3.     Hu, JW. Javaid, A. Creutzig, F. Leverage points for accelerating adoption of shared electric cars: Perceived benefits and environmental impact of NEVs. ENERGY POLICY, 2021, 155, 112349.

4.     Kylili, A; Afxentiou, N; Georgiou, L; Panteli, C; Morsink-Georgalli, PZ; Panayidou, A; Papouis, C; Fokaides, PA. The role of Remote Working in smart cities: lessons learnt from COVID-19 pandemic. ENERGY SOURCES PART A-RECOVERY UTILIZATION AND ENVIRONMENTAL EFFECTS, 2020, DOI: 10.1080/15567036.2020.1831108

5.     Nizetic, S. Impact of coronavirus (COVID-19) pandemic on air transport mobility, energy, and environment: A case study. INTERNATIONAL JOURNAL OF ENERGY RESEARCH, 2020, 44, 10953-10961.

6.     Schulte-Fischedick, M; Shan, YL; Hubacek, K. Implications of COVID-19 lockdowns on surface passenger mobility and related CO2 emission changes in Europe. APPLIED ENERGY, 2021, 300, 117396.

7.     Lalas, D; Gakis, N; Mirasgedis, S; Georgopoulou, E; Sarafidis, Y; Doukas, H. Energy and GHG Emissions Aspects of the COVID Impact in Greece. ENERGIES, 2021, 14, 1955.

8.     Chiaramonti, D; Maniatis, K. Security of supply, strategic storage and Covid19: Which lessons learnt for renewable and recycled carbon fuels, and their future role in decarbonizing transport? APPLIED ENERGY, 2020, 271, 115216.

The results of linear regression should be discussed in more detail.

Conclusions section needs to be improved by highlighting the findings of the research.

Also, I recommend to add the limitations of the study in the Conclusions section.

Author Response

We thank the reviewer for their feedback! Please see the attached file where we address each comment one by one.

Reviewer 2 Report

This article presented a detailed exploration of people activities before, during and after the COVID-related lockdown in Austria. The research results revealed the impacts of the lockdown on people’s lives and mobilities. I recommend a minor revision. 

(1) Due to the scope of this journal, the authors should discuss their research motivations and considerations from the perspective of sustainable development. 

(2) In Introduction, the authors should elaborate the research gaps and the scientific questions. 

(3) The term Secondary Activities should be explained at its first mention.

 (4) In figure 2 and the corresponding text, it will be better to present the proportion of trips for each travel mode.

 (5) In figures 3 and 4, only Shopping at home decreased during the lockdown compared to before. The change percentage is negative in figure 4. The authors should discuss it more.

 (6) In table 1 and 2, the authors selected different moderator variables for the same dependent variable. For Sleep, for example, three moderator variables were used in table 1, but eight variables in table 2. Why? In addition, different dependent variables were explained by different moderator variables. Why can’t the dependent variables be explained using the same set of moderator variables? What are the criteria for selecting moderator variables?

Author Response

(The authors gave the same response as above.)

Round 2

Reviewer 1 Report

The revised paper has taken into consideration my suggestions. In my opinion, the article is well-written and is worth to be published.